# Coarse-to-Fine Localization for Detecting Misalignment State of Angle Cocks

**DOI:** 10.3390/s23177311

**Published:** 2023-08-22

**Authors:** Hengda Lei, Li Cao, Xiuhua Li

**Affiliations:** 1School of Electrical and Electronic Engineering, Wuhan Polytechnic University, Wuhan 430023, China; 2Wuhan Huamu Information Technology Co., Ltd., Wuhan 430070, China

**Keywords:** angle cock, non-closed state, misalignment state, handle localization

## Abstract

The state of angle cocks determines the air connectivity of freight trains, and detecting their state is helpful to improve the safety of the running trains. Although the current research for fault detection of angle cocks has achieved high accuracy, it only focuses on the detection of the closed state and non-closed state and treats them as normal and abnormal states, respectively. Since the non-closed state includes the fully open state and the misalignment state, while the latter may lead to brake abnormally, it is very necessary to further detect the misalignment state from the non-closed state. In this paper, we propose a coarse-to-fine localization method to achieve this goal. Firstly, the localization result of an angle cock is obtained by using the YOLOv4 model. Following that, the SVM model combined with the HOG feature of the localization result of an angle cock is used to further obtain its handle localization result. After that, the HOG feature of the sub-image only containing the handle localization result continues to be used in the SVM model to detect whether the angle cock is in the non-closed state or not. When the angle cock is in the non-closed state, its handle curve is fitted by binarization and window search, and the tilt angle of the handle is calculated by the minimum bounding rectangle. Finally, the misalignment state is detected when the tilt angle of the handle is less than the threshold. The effectiveness and robustness of the proposed method are verified by extensive experiments, and the accuracy of misalignment state detection for angle cocks reaches 96.49%.

## 1. Introduction

In the field of railway transportation, fault detection for key components in the braking system of freight trains is critical for ensuring railway transportation safety [1] since the abnormal state of these components can result in serious consequences. Therefore, it is very necessary to detect the abnormal state of these components of freight trains during transportation so as to deal with these anomalies in time to ensure the safety of freight trains [2]. The traditional method of fault detection is mainly carried out by manual work. The train inspection personnel determine whether each component is in the abnormal state by means of ‘‘touch, look, and listen’’ [3]. The efficiency of manual detection depends entirely on the working state of the inspectors, and the subjectivity is pretty strong [4]. In order to improve the efficiency and accuracy of fault detection, the methods based on image processing and machine vision are gradually applied in the field of fault detection for key components in the braking system of freight trains [5].

As one of the key components in the braking system, the angle cock is the switch of the train ventilation duct, which realizes the braking through the air transmission between carriages, so that the train can achieve the purpose of regulating speed or stopping [6]. If the state of angle cock is abnormal, the train will be unable to brake normally, which will lead to overrunning, rear-end collision, etc. For an angle cock, there are three steps in the process of changing it from the closed state to the fully open state, namely lifting the handle of angle cock upward, rotating its handle to the left, and pressing its handle downward. Once the train staff misses the process of pressing its handle down, the angle cock will be in another state other than the closed and fully open state, which is called the misalignment state. If the angle cock is in the fully open state, the air in the ventilation duct flows normally. If the angle cock is in the closed state, the air in the ventilation duct cannot flow properly. But if the angle cock is in the misalignment state, the air will leak into the external environment during transmission, which leads to the reduction of air in the ventilation duct and prevents normal air circulation. Therefore, the misalignment state, along with the closed state, belongs to the abnormal state. In the fault detection of angle cocks, it is necessary to detect not only the closed state, but also the misalignment state. The existing methods equate the non-closed state (which includes the fully open state and the misalignment state) to the fully open state and implement the fully open and closed state detection. For example, the CNN-based detector called Light FTI-FDet model, proposed by Zhang et al. [7], can only implement the fully open and closed state detection for angle cocks, though it can achieve multi-fault detection for freight trains. The real-time and accurate fault inspection approach proposed by Zhou et al. [8] is only used to detect the missing handle of an angle cock. Apparently, the misalignment state is ignored or misclassified as normal. Therefore, it is essential to further identify the misalignment state from the non-closed state. 

At present, considering that the state of angle cocks is obtained by analyzing and processing the image of angle cocks, the handle of an angle cock is the decisive factor. In this paper, we propose a coarse-to-fine localization method for detecting misalignment state of angle cocks through detecting their handle state, where the coarse localization is realized by using the you only look once version 4 (YOLOv4) model and the fine localization was achieved by using a support vector machine (SVM) model.

The whole process of this method is divided into two steps: localization and detection. In the localization step, the camera is installed in the wilderness. Therefore, the image quality will be affected by environmental factors such as weather and light [9]. Since the camera installation location is not fixed, the angle cock cannot be localized by the relative location relationship between itself and other parts [10]. To reduce the computation, we propose a coarse-to-fine method based on YOLOv4 model and SVM model to locate the handle of an angle cock. Meanwhile, the histogram of oriented gradient (HOG) features was selected as image features. Although the camera has multiple optional installation positions, once the installation position is selected, the position of angle cocks on each train is also fixed. Therefore, the captured image will not be rotated or offset [11]. In the detection step, the closed and non-closed state of an angle cock is distinguished by using the SVM model combined with the HOG feature of its handle localization result, and the curve fitting of the handle is achieved by window search. And the tilt angle of the minimum bounding rectangle is calculated and compared with the threshold. The contributions of this paper are mainly as follows:A coarse-to-fine method based on the YOLOv4 and SVM model is proposed to locate the handle of an angle cock, which is accurate and has small computations.The state of angle cocks is detected only by their handles, which reduces interference in the localization results.A window search method is used for the curve fitting of the handle, which has good effect and stable performance.

## 2. Related Works

In the fault detection of freight trains, most of the studies based on image processing and machine vision extract relevant image features to determine whether the component is in the abnormal state. The common image features include two categories: human-designed features and deep features. The basic image features such as shape, texture, and fusion features belong to the human-designed features. And the features obtained by training with deep learning models belong to the deep features [12]. For the fault detection of various components, suitable feature representations need to be selected according to their own characteristics, and specific fault judgment guidelines should be used [13].

For human-designed features, Liu et al. [13] used sparse histograms of oriented gradients (SHOG) combined with a SVM model to identify the fault of bogie block key with an accuracy of 99.86%. Liu et al. [14] proposed a method that combines gradient coding co-occurrence matrix (GCCM) and AdaBoost to detect the state of bogie block key with an accuracy above 99.6%. Li et al. [15] used the local binary patterns (LBP) and a SVM model to judge whether the bolts were in the normal state. The accuracy and false alarm rate were 100% and 8.1%, respectively. Liu et al. [16] achieved the state detection of fastening bolts through gradient orientation co-occurrence matrix (GOCM) combined with a SVM model, whose accuracy reached 99.91%. Cha et al. [17] used Hough transform and the height of bolts to represent two states of looseness and tightness. Following that, a linear support vector (LSVM) was used to detect which state the bolts were in, and the accuracy reached 95.45%. Zheng et al. [18] combined the gradient magnitude and the histogram of gradients in six directions to form the multi-dimensional features (MDF) to locate the coupler yoke with a SVM model. Following that, Haar-like features and AdaBoost decision trees were selected to detect the loss of bolts with an accuracy of 98.60% and a false alarm rate of 4.1%. The human-designed features require different feature extraction algorithms for different targets. And it is difficult to find the most suitable features to characterize the targets, which leads to being unable to detect multiple targets at the same time. 

For deep features, since the deep learning models can adaptively extract optimal features and detect multiple targets at the same time, they are widely used in the field of fault detection. Sun et al. [19] used convolutional neural networks to achieve coarse localization of end bolts and side keys. Next, the precise localization was accomplished through the prior information, geometric and spatial location relationships, and another neural network was used to achieve fault detection. The accuracy of fault detection for these two components was 97.50% (when the bolts of shaft bolts are missing), 92.5% (when the bolts of shaft bolts are loose), 100% (when the shaft bolts are missing), and 100% (when the side frame keys are missing). Ye et al. [20] proposed a multi-feature fusion network (MFF-net), in which the multi-branch dilated convolution module (MDCM), squeeze, and excitation block (SEB) were added to achieve fault detection of side bolts, bottom bolts, and the retaining key. Finally, the mean average precision reached 92.55%, 97.18%, and 97.41%, respectively. Zhang et al. [7] proposed a CNN-based Light FTI-FDet, using multi-region proposal network (MRPN) and model reduction scheme to achieve fault detection of angle cock, bogie block key, dust collector, cut-out cock, and fastening bolt. Finally, the accuracy reached 100%, 99.86%, 99.53%, 96.24%, and 100%, respectively. Ye et al. [21] used a multi-mode aggregation feature enhanced network (MAFENet) based on a single-stage detector (SSD) to achieve detection of side bolts, bottom bolts, and the retaining key. Finally, the mean average precision (mAP) reached 97.51%, 97.88%, and 100%, respectively. The deep features require a large amount of data for training, which can extract deep and abstract features of the target. The detection performance is less affected by the image environment factors, while the drawback is complex and computationally intensive.

In terms of the research object of this paper, the current research focuses on the fully open and closed state detection of angle cocks. Zhang et al. [7] used a Light FTI-FDet to solve the fully open and closed state detection with an accuracy of 100%. This method requires tuning a large number of parameters and performing multiple experiments to obtain the optimal network, which consumes a lot of computing resources. Meanwhile, the obtained model is complex due to the massive modifications for the network. Zhou et al. [8] used gradient encoding histograms (GEH) combined with a SVM model to achieve the state detection of angle cocks with an accuracy of 99.80%. This method requires a great deal of computation on the image of angle cocks to extract the features of angle cocks. Since the handles and buckles of angle cocks are involved in the calculations, while the handles are the key factor, it involves a lot of unnecessary calculations. In addition, these two methods only realize the detection of fully open and closed state of angle cocks. 

To solve these problems, this paper proposes a method for misalignment state detection of angle cocks. Firstly, the closed and non-closed states are distinguished by a SVM model on the basis of the coarse-to-fine localization result. After obtaining the detection result of the non-closed state, the misalignment state, if present, can be detected by setting an appropriate angle threshold for the handle of an angle cock. The whole detection process is realized only by detecting the state of the handle of an angle cock and does not include complex network structure, so the calculation is relatively simple. What is more, the proposed method can be used to precisely detect three different states of angle cocks, especially the ignored misalignment state.

## 3. Method

The algorithm framework is shown in Figure 1, which is mainly divided into four parts. In the first part, a YOLOv4 model is used to obtain the localization result of an angle cock from the original image (including the handle and buckle of the angle cock). In the second part, the sub-images are generated in the vertical direction with a fixed size image, and the HOG features are extracted and inputted into to the SVM model to obtain the localization result in the vertical direction. Subsequently, the same operation is performed in the horizontal direction to obtain the handle localization result of the angle cock. In the third part, the HOG features and SVM model continue to be applied to the handle localization result to detect the closed and non-closed state. In the fourth part, if the angle cock is in the non-closed state, the binarization of the handle is performed, and the window search is used to fit the handle curve. The minimum bounding rectangle of the curve and the tilt angle of the rectangle is calculated. Finally, the tilt angle is compared with the threshold to determine whether the angle cock is in the misalignment state or not. 

### 3.1. YOLOv4 Model

The YOLOv4 model is widely used in the field of target detection and is able to locate the target with complex backgrounds and a great deal of noise [22]. The class and location of target can be obtained through a large amount of data training [23]. As the angle cocks appear on the mutual hook stations, many other components in these stations can cause interference to the localization of angle cocks. Therefore, it is essential to ensure that the localization result is not affected by other components, the background environment [23], the position changing of the angle cocks caused by multiple optional installation positions of camera, and the difference of sample data caused by different time, weather, and train models. To solve these problems, the YOLOv4 model was selected for the localization of angle cocks.

### 3.2. HOG Feature

The HOG feature is a gradient-based feature. It can describe the appearance or shape of an object by using the gradient or density distribution in the edge direction of the image, which can reduce the impact of light and shadow [24]. Since the image of angle cocks is taken from a different environment, the image quality will be affected by many factors such as illumination and clarity [25]. The HOG feature contains the process of compressing the illumination and the edge [26], it can solve the problem of large change in the illumination and background contrast. The different states of the angle cock have obvious difference in the image gradient, so the HOG feature was selected.

### 3.3. SVM Model

Using the YOLOv4 model can only obtain the localization result of an angle cock, which is difficult to identify its handle and thus extract the tilt angle of the handle for detecting the misalignment state of the angle cock. Therefore, the SVM model in the vertical and horizontal direction should be used to locate the handle of an angle cock [27]. The SVM is a widely used binary classification algorithm which can predict classification results for any input. The SVM model is suitable for solving the problems of small sample, nonlinear, and high-dimensional pattern recognition [28]. In the field of fault detection, the states of a component can be divided into normal and abnormal states. Different states have different image features. For the state detection of angle cocks, whether the handle of an angle cock is included in the sub-images or not, and whether it is in the closed state or non-closed state, belongs to the two-class problem [29]. Therefore, the SVM model was selected here as a discriminator for the former case and a classifier for the latter case.

### 3.4. Window Search

Window search refers to scanning the image by preset width and height of the window [30], searching the entire image with a fixed size window. During the search process, the search results can be improved by changing the number of blocks in the vertical and horizontal directions. The target information of each window can be stored until the end of the search [31]. The specific process of window search is displayed in Figure 2, where $O_{n}$ represents the *n*-th search starting point, while $L_{n}$ is defined as the *n*-th fitting point.

Firstly, a rectangular window with width of w and height of h in the upper left corner of the image is selected, and its upper left corner is regarded as the search starting point, which is denoted as $O_{1}$. The white pixel area in the window is projected vertically to obtain the abscissas of all white pixel points. Meanwhile, the median of these abscissas is calculated as the abscissa of the fitting point ($L_{1}$), expressed as $x_{1}$. In each rectangular window, the proportion of white pixels is random and different. Therefore, the value of abscissa of the fitting point ($x_{1}$) is not necessarily half of w, while the abscissa of the next search starting point ($O_{2}$) is half of w. At the same time, the ordinate of the fitting point is taken as half of h. Therefore, the coordinate of the first fitting point $L_{1}$ is $\left(x_{1},\frac{h}{2}\right)$. This rule traverses over the entire image to obtain the coordinates of the starting point ($O_{i},i\in \{1,2,\ldots ,n\}$) and the fitting point ($L_{i}$) of each search. Following that, the coordinates of each obtained fitting point ($L_{i}$) can be stored until all rows or columns of the entire image are traversed. When the search is over, all the stored fitting points are connected to form the final fitting curve [32].

## 4. Experiments and Results

The experiments conducted in this paper were performed on a PC with a 12th Gen Inter Core (R) Core (TM) i7-12700 2.10 GHz CPU, 32G RAM, and a graphics card Inter (R) UHD Graphics 770 (Santa Clara, CA, USA). The software environment is based on Windows 11, Pytorch 1.7.0, Python 3.10.9, CUDA 11.0, and cuDNN 8.0.5.

All the images containing different states of angle cocks were obtained from the railway department to ensure the reliability of data. For the experiment of the localization of angle cocks in Section 4.1, 1841 images were used for training, and 789 images were used for testing. For the rest of the experiments in Section 4.2, Section 4.3 and Section 4.4, 624 images randomly selected from the training and testing data were applied to realize the localization of handles, the misalignment state detection, and robustness tests, respectively. 

### 4.1. The Localization of Angle Cocks

The YOLOv4 model was used for the localization of angle cocks. For each localization result, it includes the handle and the buckle, as shown in the red ellipse and rectangular box in Figure 3. The result contains both a label and a purple rectangular box, where the label represents the detection accuracy and the purple rectangular box reflects the localization information, which can be used to cut out the angle cock from the original image.

### 4.2. The Localization of Handles

After obtaining the localization results of angle cocks, it is still essential to locate the handle of each angle cock due to the difficulty of detecting the misalignment state of the angle cock through extracting its whole external contour information. The accuracy of the handle localization depends mainly on the following two parameters:Sliding step: the original image size is fixed when generating sub-images in the vertical and horizontal directions. A variable parameter is the sliding step in both directions, which determines the quantity and similarity of sub-images.Image features: after obtaining the sub-images in the vertical and horizontal directions, the appropriate image features are selected and inputted into the SVM model for training. Different features can affect the classification results of the SVM model.

In the selection of evaluation criteria, the following three evaluation metrics are used: accuracy (*Acc*), precision, and recall, which can be calculated by the following Equation (1), Equation (2), and Equation (3), respectively.
(1)$$Acc=\frac{TP+TN}{TP+TN+FP+FN}\times 100\%$$
(2)$$Precision=\frac{TP}{TP+FP}\times 100\%$$
(3)$$Recall=\frac{TP}{TP+FN}\times 100\%$$
where *TP* stands for the number of true positive cases, *FP* stands for the number of false positive cases, *FN* stands for the number of false negative cases, and *TN* stands for the number of true negative cases.

#### 4.2.1. Sliding Step

Since the image features have not been determined, it is difficult to use the optimal image features to conduct sliding step experiments. Therefore, the gray images were chosen for experiments with sliding steps of different sizes to obtain the optimal value of sliding step.

In the vertical direction, the image size of localization result for each angle cock is 150 × 150. Therefore, the vertical and horizontal fixed image sizes are 150 × 75 and 75 × 75, respectively. Experiments were conducted with step sizes of 5, 15, and 25. Finally, the results and P-R curves are listed in Table 1 and Figure 4, respectively.

As can be seen from Table 1, when all sub-images are gray images, the accuracy increases continuously with the sliding step in both vertical and horizontal directions. Similarly, it can be seen from the P-R curves in Figure 4 that the area in both vertical and horizontal directions is highest when the sliding step is set as 25, where the area represents the mAP. Therefore, the image sliding step was set as 25 for both vertical and horizontal directions.

#### 4.2.2. Image Features

In the case of a fixed sliding step, different image features can affect the accuracy of SVM models for the handle localization. Therefore, the comparison experiment is conducted with different image features, which include the Gray [14], HOG, and LBP features [15]. The experiment result is shown in Table 2 and Figure 5. It is worth noting that the sliding step of images is taken as 25 here.

As seen in Table 2, when the direction remains the same, the accuracy of handle localization by using the HOG feature is the highest, which can reach 98.29% in the vertical direction and 98.06% in the horizontal direction. Both of them are higher than those using the Gray and LBP features. Similarly, it can be seen from the P-R curves in Figure 5 that the area of the P-R curves by using the HOG feature is larger compared to the other two features, where the area represents the mAP.

For object detection, it is very important to find the edge information of the target. The Gray feature is simply a grayscale of the original image and does not contain gradient information. While the HOG and LBP features contain gradient information, the edge points of the target can be easily obtained. Furthermore, considering that the image background of angle cocks is affected by the weather and time but the direction of angle cocks is invariant, the LBP feature descriptor suitable for extracting targets with variable directions is inferior to the HOG feature descriptor suitable for extracting targets with a high interference from the background. In this way, the HOG feature is chosen.

Through the analyses of the above two variable parameters, it can be determined that the sliding step is set as 25 and the HOG feature is selected for the image feature. The handle localization is implemented with the SVM model based on the localization result of angle cocks. 

Firstly, the sub-images are made in the vertical direction in Figure 6a. The size of each sub-image is fixed as 150 × 75 and the sub-images are generated by sliding in steps of 25 from top to bottom. One of them is depicted in Figure 6b. The HOG features of the sub-image in Figure 6b are illustrated in Figure 6c. All the HOG features of sub-images in vertical direction are extracted and inputted to a SVM model as training sample to obtain the localization result. Similarly, for each sub-image in vertical direction, the same operations are performed in horizontal direction, where the size of each sub-image is reduced to 75 × 75 and the step size is set as 25 from left to right. Finally, based on the localization result of an angle cock, the handle localization result can be obtained using the SVM model, as presented in the red rectangular box in Figure 6a.

### 4.3. Misalignment State Detection

After obtaining the handle localization result of an angle cock, the non-closed state can be detected firstly. The HOG features are used as two different feature forms for the closed and non-closed state, which are then judged by a SVM model. If the angle cock is in the non-closed state, then the misalignment state should be detected. This detection is realized by determining whether the tilt angle of the minimum bounding rectangle for the handle exceeds the threshold. Before performing the curve fitting operation, the obtained handle of an angle cock was first binarized and extracted by using connected domain analysis, as shown in Figure 7.

After obtaining the target handle, the curve fitting operation was performed by the window search. In the process of window search, there are four variable parameters: numx, numy, w, and h, where w and h represent the width and height of the window, respectively. The value of w depends on numx and the image width of localization result for each handle, while the value of h depends on numy and the height of localization result for each handle, where the width and height is fixed. Therefore, the curve fitting operation depends mainly on the following two parameters:numy: the number of vertical block;numx: the number of horizontal block.

#### 4.3.1. Numy

Assuming the value of numx is 8 and remains constant. The optimal value of numy is determined by changing numy and observing the result of the curve fitting operation, as shown in Figure 8.

As can be seen from Figure 8, the curve starts getting close to the handle when numy≥16. When numy>24, the fitting curve gradually deviates from the handle. With the increase of numy, the fitting curve becomes more and more complete until it can be fully extended to the entire handle. In summary, the optimal value of numy is 24.

#### 4.3.2. Numx

The value of numy was set as the optimal value unchanged and the optimal value of numx was determined by changing numx and observing the result of the curve fitting operation, as presented in Figure 9.

Figure 9 illustrates that the curve gradually fits the handle when numx≥6. When numx>10, the middle of the fitting curve gradually becomes deformed. When numx increases from 6 with an interval of 2, the curve changes minimally. In summary, the optimal value of numx is 8.

By determining the optimal values of these two adjustable parameters in the window search, the curve of the handle can be obtained. Following that, the optimal fitting curve is extracted, as depicted in Figure 9c. The minimum bounding rectangle of this curve is shown in Figure 10a. Next, the tilt angle of this rectangle was calculated, which refers to the angle between the rectangle and the horizontal axis. Finally, the tilt angle is displayed in Figure 10b.

After the angle value of the handle was obtained, the misalignment state detection was performed. The comparison experiments are conducted by setting different angle thresholds, and the results are shown in Table 3.

As can be seen from Table 3, when the angle threshold is gradually increased from 41 to 45, the detection accuracy of the misalignment state increases with the increase of threshold. When the threshold continues to increase, the accuracy decreases. When the angle threshold is set as 45, the accuracy is the maximum. Therefore, it can be determined that the angle threshold of 45 is sufficient for the misalignment state detection. When the angle of the handle is less than 45, the angle cock is in the misalignment state, otherwise, the angle cock is in the fully open state. The final experimental result shows that the detection accuracy of the misalignment state is 96.49%.

### 4.4. Robustness Tests

In order to verify the robustness of this method, the salt and pepper noise (SPN) with different signal to noise ratios (SNR), along with the Gaussian noise (GN) with different variances (VAR) of mean 0, and the Poisson noise (PN) with different wavelengths, were added to the original image to represent the images acquired in different environments. The test result is shown in Figure 11. As long as the image quality is changed, the accuracy of the misalignment state detection fluctuates slightly. The SPN appears as a discrete distribution of pure white or black pixel dots on the image. These random pixel dots cover some information of the angle cocks, which leads to the greatest effect on the detection of angle cocks. When the SNR is 0.05, the detection accuracy decreases by 3.51% compared to the normal image. As the SNR continues to increase, the detection accuracy decreases. The GN is an additive noise. When the VAR is 0.01, the detection accuracy decreases by 2.63%, and as the VAR increases, the detection accuracy gradually decreases. For the PN, the detection accuracy decreases by 2.63% when the wavelength is taken as 0.01. As the wavelength increases, the detection accuracy also decreases. The reason for this is that the noises blur the contour of the target and decrease the target detection accuracy. This experiment result shows that our method has strong robustness to the images in different noisy environments.

## 5. Conclusions

This paper proposes a method for the misalignment state detection of angle cocks, with the detection accuracy of 96.49%. The localization area, only including the handle of an angle cock, was obtained from coarse to fine localization by using the YOLOv4 and SVM model, which greatly reduces the redundant calculations and provides a stable and clear sub-image for the state detection of angle cocks. Once the angle cock is judged to be in the non-closed state by using the HOG features and SVM model on the handle localization result, the handle curve is obtained by using binarization and window search, and the misalignment state detection is performed by comparing the tilt angle of the handle curve with a threshold. The experiments on the values of relevant parameters show that the proposed method can achieve a high localization and detection accuracy by setting appropriate parameter values, and the robustness tests verify the feasibility of this method. Further research will focus on finding other effective operations or models to improve the accuracy and generalization performance for the misalignment detection of angle cocks.

## Figures and Tables

**Figure 1 sensors-23-07311-f001:**
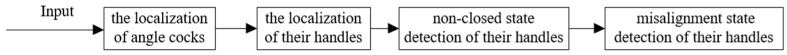
Algorithm framework.

**Figure 2 sensors-23-07311-f002:**
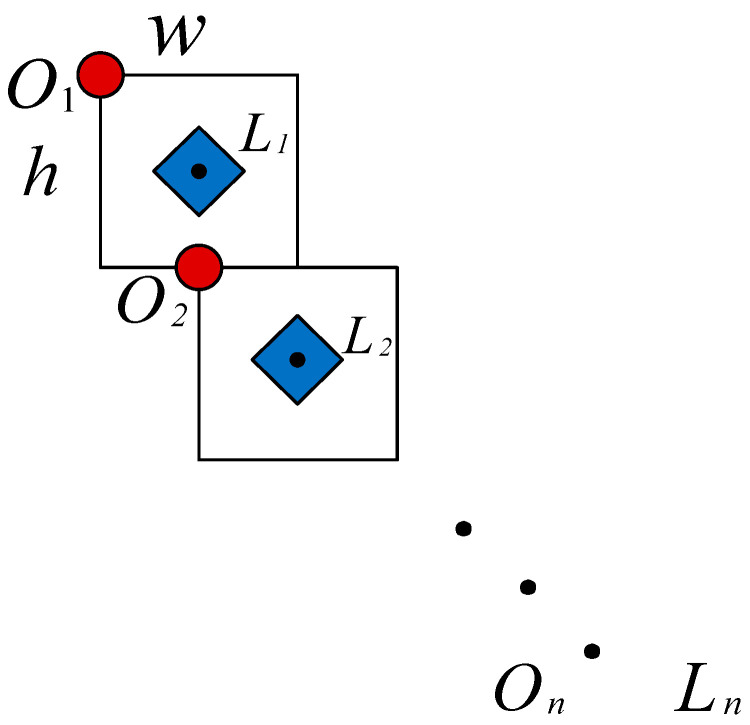
Window search.

**Figure 3 sensors-23-07311-f003:**
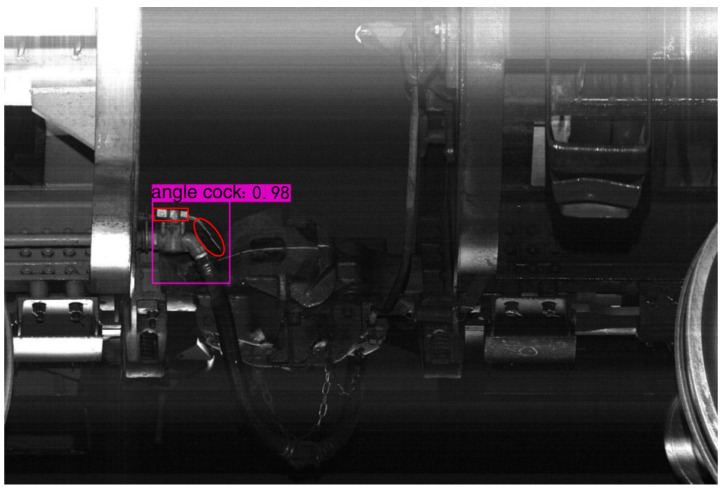
The localization of an angle cock.

**Figure 4 sensors-23-07311-f004:**
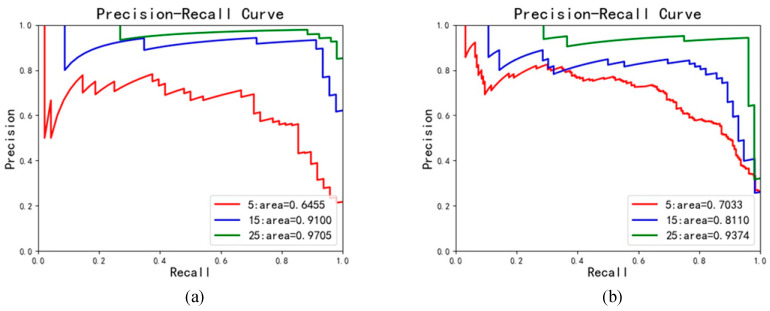
The P-R curves: (**a**) vertical direction, (**b**) horizontal direction.

**Figure 5 sensors-23-07311-f005:**
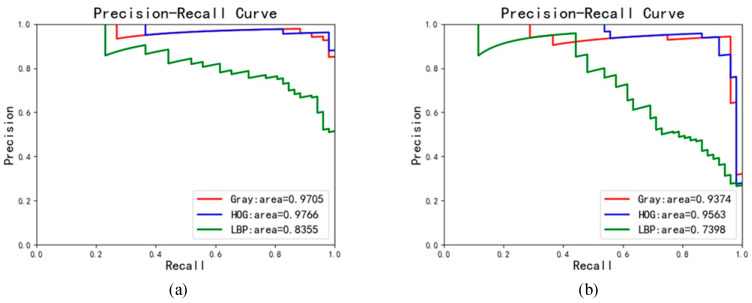
The P-R curves: (**a**) vertical direction, (**b**) horizontal direction.

**Figure 6 sensors-23-07311-f006:**
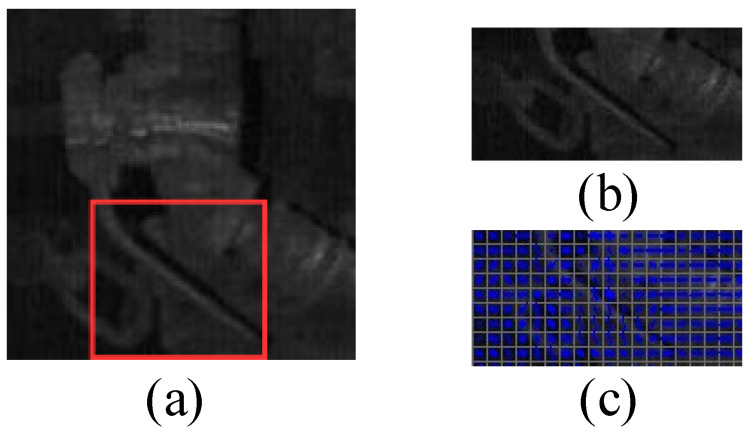
The handle localization: (**a**) the localization result of an angle cock, (**b**) a sub-image in vertical direction, (**c**) HOG features of (**b**).

**Figure 7 sensors-23-07311-f007:**
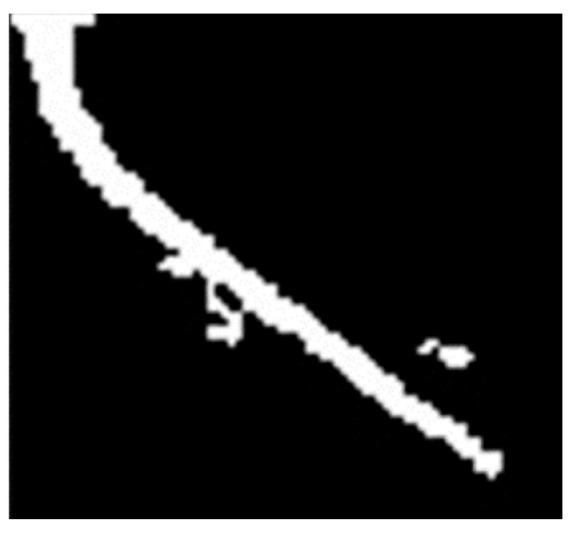
Connected component analysis result.

**Figure 8 sensors-23-07311-f008:**
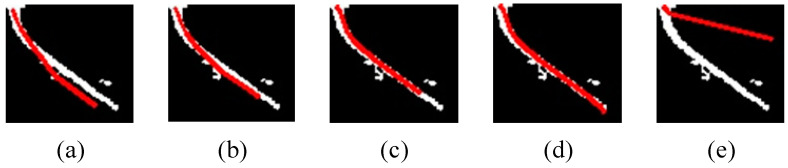
The result of curve fitting operation for different values of numy: (**a**) numy = 12, (**b**) numy = 16, (**c**) numy = 20, (**d**) numy = 24, (**e**) numy = 26.

**Figure 9 sensors-23-07311-f009:**
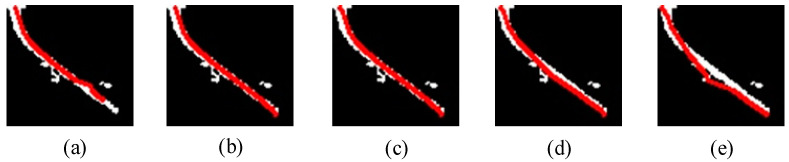
The result of curve fitting operation for different values of numx: (**a**) numx = 4, (**b**) numx = 6, (**c**) numx = 8, (**d**) numx = 10, (**e**) numx = 12.

**Figure 10 sensors-23-07311-f010:**
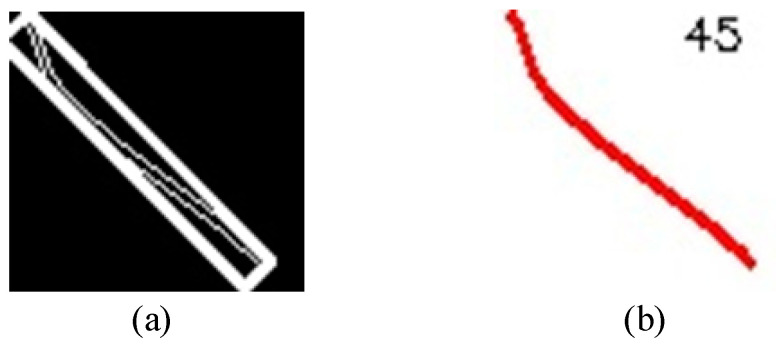
The misalignment state detection: (**a**) minimum bounding rectangle, (**b**) tilt angle.

**Figure 11 sensors-23-07311-f011:**
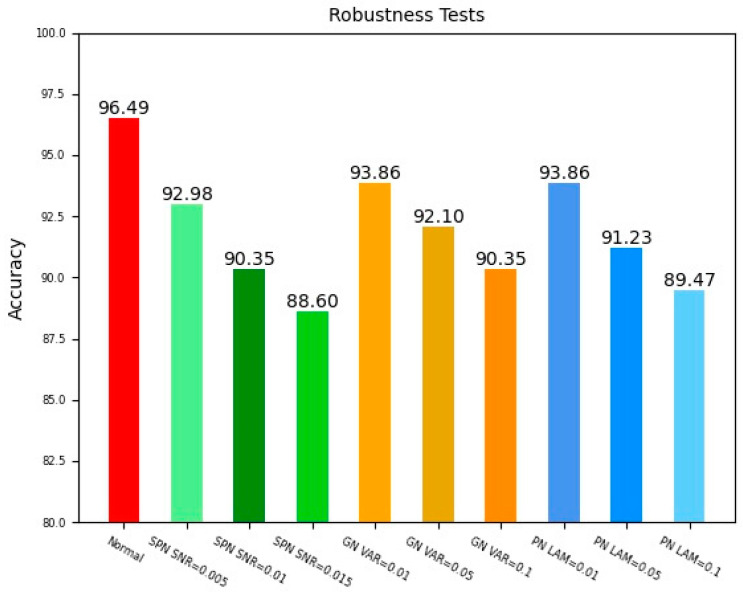
Comparison of accuracy for different noise added in the original image.

**Table 1 sensors-23-07311-t001:** Comparison of accuracy for different sliding steps.

Sliding Step	*Acc* in Vertical Direction/%	*Acc* in Horizontal Direction/%
5	94.23	76.72
15	94.89	93.27
25	97.04	97.60

**Table 2 sensors-23-07311-t002:** Comparison of accuracy for different image features.

Image Features	*Acc* in Vertical Direction/%	*Acc* in Horizontal Direction/%
Gray	97.04	96.15
HOG	98.29	98.06
LBP	80.47	84.13

**Table 3 sensors-23-07311-t003:** Comparison of accuracy for different angle thresholds.

Angle_Threshold	*Acc*/%
41	82.46
43	92.98
45	96.49
47	87.72
49	77.19

## Data Availability

The data in this study are owned by the research group and will not be transmitted.

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
