# Peer review of "Coarse-to-Fine Localization for Detecting Misalignment State of Angle Cocks"

_sensors, 2023, doi:10.3390/s23177311_

Round 1
Reviewer 1 Report
Traffic safety issues are very important and require special attention. Therefore, the topic of the article is relevant.
However, there are comments to it.
1. Lines 63 – 65 declare the results of the research, but should emphasize its relevance. I invite the authors to correct this proposal or delete it.
2. Analysis of research in this area needs to be strengthened. Here it is necessary to make the critical analysis clearer in order to highlight the previously unresolved problem.
3. The authors need to state the purpose and objectives of the study.
4. Fig. 3 is very fuzzy.
5. Formulas (1) – (3) should be referenced in the text of the article. If these are author's formulas, then this must be indicated.
6. According to the text of the article, it is necessary to provide information about the measuring equipment that was used in the experiment.
7. Fig. 6 is missing.
8. It is necessary to indicate the number of measurements taken, the reliability of the sample, and also provide a test plan.
9. Subsection 4.4 is very succinct and needs to be improved.
10. The Discussion section is very weak and needs to be improved.
11. There are no conclusions to the article.
Author Response
Response to Reviewer 1 Comments
Point 1: Lines 63 – 65 declare the results of the research, but should emphasize its relevance. I invite the authors to correct this proposal or delete it.
Response 1:Thank you very much for pointing out the problem. In the revised manuscript, the declaration of the results is divided into two sentences, and the relevance expression is carried out respectively. It is reflected in Lines 66-71 (Page 2) to emphasize the specific implementation method of the coarse-to-fine location, and the sentence about the effectiveness and feasibility has been deleted for the logic of the overall description of the article.
Point 2:Analysis of research in this area needs to be strengthened. Here it is necessary to make the critical analysis clearer in order to highlight the previously unresolved problem.
Response 2:Thank you for your suggestion. According to your suggestion, we have made improvements in two places to emphasize the analysis of research in this area. In the revised manuscript, Lines 59-64(Page 2) illustrate the idea of the existing method by citing examples from the references, emphasizing the currently unsolved problem that the misalignment state is ignored and misclassified as normal. At the same time, Lines 139-148(Page 3) provide a critical analysis of this area of angle cocks, indicating the limitations and shortcomings of existing methods. The existing methods only realize the detection of fully open and closed state of angle cocks, while ignoring the misalignment state. These critical analyses are used to highlight current unresolved issues.
Point 3:The authors need to state the purpose and objectives of the study.
Response 3:Thank you for bringing the issue to our attention. Regarding this issue, we have added an explanation of the purpose and objectives of the study. In the revised manuscript, Lines 151-159(Page 4) illustrate the purpose of the study, which is to address the problems of ignoring the misalignment state and existing a lot of redundant calculations. The goal is to precisely detect three different states of angle cocks, especially the ignored misalignment state with a high accuracy.
Point 4:Fig. 3 is very fuzzy.
Response 4:Thank you very much for pointing out the problem. Figure 3 shows the result after YOLOv4 localization without any preprocessing. The original images come from different environments and most of them are collected in the dark environment. Meanwhile, the original image is too large and cannot be inserted into Word at their original size. Therefore, this figure needs to be properly resized for editing in Word, which leads to that the figure is slightly blurred.
Point 5:Formulas (1) – (3) should be referenced in the text of the article. If these are author's formulas, then this must be indicated.
Response 5:We are extremely grateful to reviewer for pointing out this problem. These formulas are common performance evaluation metrics in this research field and have been cited on Lines 265-266 (Page 7) in the revised manuscript.
Point 6: According to the text of the article, it is necessary to provide information about the measuring equipment that was used in the experiment.
Response 6: We deeply appreciate the reviewer’s suggestion. According to the reviewer’s comment, we have added a more detailed interpretation about the experimental equipment. In the revised manuscript, Lines 234-237 (Page 6) present information about the equipment used in the experiment, mainly including the CPU, RAM, and software environment.
Point 7: Fig. 6 is missing.
Response 7: I ' m sorry for that you may not be able to see the image properly. When we inserted the Figure 6, this Figure may be missing due to an oversight or some other reason. We have reinserted Figure. 6 in the revised manuscript and at the same time attach it below for reviewing again.
Figure 6. The handle localization: (a)Original image (b)sub-image (c)HOG.
Point 8: It is necessary to indicate the number of measurements taken, the reliability of the sample, and also provide a test plan.
Response 8: We are grateful for the suggestion. To be clear and in accordance with the reviewer concerns, we have added a brief description in the revised manuscript. Lines 238-243 (Page 6) describe the sources and use of the image data for each experiment.
Point 9: Subsection 4.4 is very succinct and needs to be improved.
Response 9: Thank you for the suggestion. We have added the adequate content in Section 4.4. The influence of each type of noise is analyzed, and then the change of its accuracy is discussed in Lines 391-401 (Page 11). We analyze the causes of such results to enrich the content of this part.
Point 10: The Discussion section is very weak and needs to be improved.
Response 10: Thank you for your suggestion. We have modified this expression throughout the text according to the comment. In terms of the table and figures of each experiment, some necessary analyses and discussions are given, as seen in Lines 281-285 (Page 7), Lines 295-311 (Page 8), Lines 376-383 (Page 10), and Lines 389-401 (Page 11). We change the discussion of Section 5 into a “Conclusion”, which comprehensively summarizes the research content and conclusion.
Point 11: There are no conclusions to the article.
Response 11: Thank you for underlining this deficiency. The conclusion of this paper is given in Section 5, as seen in Lines 405-417 (Page 11-12).

Reviewer 2 Report
The authors have done a good job of presenting a detailed study to understand the misalignment state of angle cocks in freight trains. The paper shows an improvement over previous work by detecting both the open/closed state and the misalignment state of these cocks. I would recommend the authors address the following points in this article to aid the readers:
1. How many images were analyzed in this study? The accuracy, precision, and recall numbers have been reported but how big was the data set?
2. In Fig. 4 and 5, the authors discuss that using HOG features is more accurate than Gray or LBP features but the latter have not been introduced in the text at all. The authors should briefly explain how Gray and LBP features work and why they are less accurate than HOG features.
3. The authors can explain how HOG, SVM and YOLOv4 models work with schematics like Fig 2. This will help the readers understand and replicate findings of this paper better.
Minor spell check required
Author Response
Response to Reviewer 2 Comments
Point 1:How many images were analyzed in this study? The accuracy, precision, and recall numbers have been reported but how big was the data set?
Response 1:Thank you for your precious comments and advice. Those comments are all valuable and very helpful for revising and improving our paper. We have revised the manuscript accordingly in Lines 238-243(Page 6), 1841 images are used for training, and 789 images are used for testing. For the rest of the experiments in Section 4.2-4, 624 images randomly selected from the training and testing data are applied to realize the localization of handles, the misalignment state detection and robustness tests respectively.
Point 2:In Fig. 4 and 5, the authors discuss that using HOG features is more accurate than Gray or LBP features but the latter have not been introduced in the text at all. The authors should briefly explain how Gray and LBP features work and why they are less accurate than HOG features.
Response 2:We are extremely grateful to reviewer for pointing out this problem. Following your valuable comments, we have introduced the related literature of Gray and LBP in Line 289 (Page 8). Meanwhile, Lines 301-308 (Page 8) briefly introduces the working principle and reason analysis of Gray and LBP. The Gray feature is simply a grayscale of the original image, and does not contain gradient information, while LBP is suitable for extracting targets with variable directions. Considering that the image background of angle cocks is affected by the weather and time, but the direction of angle cocks is invariant, the LBP feature descriptor suitable for extracting targets with variable directions is inferior to the HOG feature descriptor suitable for extracting targets with a high interference from the background.
Point 3:The authors can explain how HOG, SVM and YOLOv4 models work with schematics like Fig 2. This will help the readers understand and replicate findings of this paper better.
Response 3:Thank you very much for your valuable feedback and comments. Figure 2 illustrates the window search process used for fitting the handle curve. The relationship among HOG, SVM, and YOLOv4 models used in the proposed method can be explained in Lines 161-173 (Page 4) in Section 3.

Round 2
Reviewer 1 Report
The authors took into account the marked remarks
Please pay attention to line 182. There is a typo in the link. It needs to be fixed
Author Response
Response to Reviewer 1 Comments(Round 2)
Point 1: The authors took into account the marked remarks. Please pay attention to line 182.There is a typo in the link. It needs to be fixed.
Response 1:Thank you very much for pointing out the problem. In the revised manuscript (Round 2), we corrected the link in previous Line 182 (as seen in the current Line 187 in Page 4) so that it can jump to the relevant reference. At the same time, we checked all the links in the full paper and made necessary corrections.
In addition to the revised manuscript, we have made several minor amendments to further improve the quality of the paper. In order to express more accurately, the first amendment is to use “localization” instead of “location” to represent locating operations (as seen in the paper title and other relevant places). The second amendment is to explain the parameters in Eq. (1)-(3) more clearly (as seen in Lines 272-274 in Page 7). The third amendment is to summarize the processes of handle localization and distinguishment of closed and non-closed state more clearly and completely (as seen in Lines 17-21 in Page 1, Lines 82-84 in Page 2, Lines 212-213 in Page 5). The fourth amendment is to describe Figure 6 more clearly (as seen in Lines 317-332 in Page 9). All these minor amendments are highlighted in green. Besides, the format of all references is normalized. Thank you again for your guidance throughout this process.